# An ECG-based artificial intelligence model for assessment of sudden cardiac death risk
Lauri Holmstrom[1], Harpriya Chugh[1], Kotoka Nakamura [1], Ziana Bhanji[1], Madison Seifer[1], Audrey Uy-Evanado[1], Kyndaron Reinier [1], David Ouyang [1,2] & Sumeet S. Chugh[1,2] ✉

## Abstract

**Background** Conventional ECG-based algorithms could contribute to sudden cardiac death (SCD) risk stratification but demonstrate moderate predictive capabilities. Deep learning (DL) models use the entire digital signal and could potentially improve predictive power. We aimed to train and validate a 12 lead ECG-based DL algorithm for SCD risk assessment.

**Methods** Out-of-hospital SCD cases were prospectively ascertained in the Portland, Oregon, metro area. A total of 1,827 pre- cardiac arrest 12 lead ECGs from 1,796 SCD cases were retrospectively collected and analyzed to develop an ECG-based DL model. External validation was performed in 714 ECGs from 714 SCD cases from Ventura County, CA. Two separate control group samples were obtained from 1342 ECGs taken from 1325 individuals of which at least 50% had established coronary artery disease. The DL model was compared with a previously validated conventional 6 variable ECG risk model.

**Results** The DL model achieves an AUROC of 0.889 (95% CI 0.861–0.917) for the detection of SCD cases vs. controls in the internal held-out test dataset, and is successfully validated in external SCD cases with an AUROC of 0.820 (0.794–0.847). The DL model performs significantly better than the conventional ECG model that achieves an AUROC of 0.712 (0.668–0.756) in the internal and 0.743 (0.711–0.775) in the external cohort.

**Conclusions** An ECG-based DL model distinguishes SCD cases from controls with improved accuracy and performs better than a conventional ECG risk model. Further detailed investigation is warranted to evaluate how the DL model could contribute to improved SCD risk stratification.

## Plain language summary

Sudden cardiac death (SCD) occurs when there are problems with the electrical activity within the heart. It is a common cause of death throughout the world so it would be beneficial to be able to easily identify individuals that are at high risk of SCD. Electrocardiograms are a cheap and widely available way to measure electrical activity in the heart. We developed a computational method that can use electrocardiograms to determine whether a person is at increased risk of having a SCD. Our computational method could allow clinicians to screen large numbers of people and identify those at a higher risk of SCD. This could enable regular monitoring of these people and might enable SCDs to be prevented in some individuals.

Sudden cardiac death (SCD) is a major, global public health problem[1]. In Europe and the United States, ~700,000 individuals will suffer from this mostly lethal condition on a yearly basis[1,2]. Given the high mortality rate of SCD, effective primary prevention could make a substantial positive impact but the current approach needs augmentation[3,4]. Based on randomized clinical trials, patients identified to be at high risk based on severely reduced left ventricular systolic function (LVEF < 35%) receive implantable cardioverter-defibrillators[4,5]. However, there is no existing risk stratification methodology for individuals with LVEF > 35% that make up 70% of

community SCD[6,7]. Moreover, ~40–50% of all SCD cases occur in individuals without previously diagnosed cardiac disease, which is a prerequisite for SCD risk assessment.

Some novel prediction methodologies that extend beyond the left ventricular ejection fraction have been developed[8] but these are still in the research domain. Especially, the standard 12 lead ECG has received a lot of interest in the research field in anticipation of improving long-term SCD risk stratification[9]. Various ECG abnormalities have been identified to associate with an increased long-term risk of SCD[10–12], and we have

[1]Center for Cardiac Arrest Prevention, Department of Cardiology, Smidt Heart Institute, Cedars-Sinai Medical Center, Los Angeles, CA, USA. [2]Division of Artificial Intelligence in Medicine, Department of Medicine, Cedars-Sinai Medical Center, Los Angeles, CA, USA. ✉e-mail: sumeet.chugh@cshs.org

previously published a 6 variable ECG electrical risk score that identifies individuals at an increased risk of SCD[10]. However, conventional ECG-based risk stratification tools are usually limited by low accuracy or practicality, since they include measurements that are not part of a usual ECG interpretation, thus requiring customized measurement or trained medical personnel interpretation.

In recent years, ECG-based deep learning (DL) algorithms have been developed and are being deployed for diagnostic purposes[13]. ECG-based DL models have been successfully trained to detect various cardiac conditions, e.g., LV dysfunction[14], HCM[15] or to recognize patients at high risk for atrial fibrillation[16]. As opposed to conventional ECG analysis, DL models do not require manual selecting and extracting of relevant features, which enables them to capture the entire ECG signal and achieve higher prediction accuracy.

In the current study, we train, test, and validate an ECG-based DL model to identify individuals at high risk of SCD, and compare the predictions with a previously published and validated conventional ECG electrical risk score[10]. The model accurately distinguishes sudden cardiac death cases from controls, performing better than the conventional ECG risk score.

## Methods
### Study design
We used two geographically separate community-based, prospective, and ongoing studies of out-of-hospital SCDs in the general population: Oregon SUDS (training, validation, and testing) and Ventura PRESTO (external validation). Given that CAD is the most common underlying substrate for SCD, our control group was designed to represent a control sample with a similar prevalence of previously diagnosed CAD.

### SCD cohorts (Oregon SUDS & Ventura PRESTO)
Detailed methods for Oregon SUDS and Ventura PRESTO studies have also been published earlier[7,17,18]. Both Oregon SUDS and Ventura PRESTO studies ascertain all out-of-hospital SCDs from the Portland Oregon metro area (population ~1 million, Oregon SUDS, since 2002), and Ventura County, California (population ~850,000, Ventura PRESTO, since 2015) using an identical approach. Potential SCD cases in the community are identified in collaboration with each region's emergency medical services (EMS) system. Subsequently, established adjudication methods to confirm likely cardiac etiology of SCD were employed by trained physician-researchers; using all available medical record data for each potential SCD case, EMS prehospital care reports, medical examiner's reports, and death certificates from Oregon and California state vital statistics records. SCD was defined as a sudden loss of pulse due to a likely cardiac etiology that occurred with a rapid witnessed collapse, or if unwitnessed, the subject should have been seen alive within 24 h. We included successfully resuscitated cases in addition to non-survivors. Cases of likely non-cardiac etiology (e.g., trauma or substance abuse) or chronic terminal illness were excluded.

All cases with archived resting 12 lead ECGs available for analysis were included (Fig. 1). These ECGs were recorded prior to and unrelated to the SCD event, with a calibration of 10 mm/mV and paper speed of 25 mm/s. ECGs with paced rhythm, atrial fibrillation, or atrial flutter were excluded a priori to create a DL model that could be applied to ECGs in sinus rhythm. Pre-arrest clinical records and ECGs were available if the patient provided written consent or was deceased, in which case consent was waived by the institutional review boards. Institutional review boards of Ventura County Medical Center, Oregon Health and Science University, Cedars-Sinai Health System, and all other relevant health systems and participating hospitals approved the study protocol.

### Control population
We recruited control subjects from the Portland Oregon metro area to represent individuals at intermediate risk of SCD with a large proportion having CAD. Institutional review boards of Ventura County Medical Center, Oregon Health and Science University, Cedars-Sinai Health System,

and all other relevant health systems and participating hospitals approved the study protocol, and all control subjects gave informed consent for their data to be used in the study. Control subjects were identified through multiple sources, including patients undergoing angiography, patients having their chest pain assessed by EMS, or patients visiting an outpatient cardiology clinic. We ascertained the control subjects so that the prevalence of CAD and MI was comparable to SCD cases. Control patients had no previous history of cardiac arrest or ventricular arrhythmias. Matching cases and controls for underlying CAD enables the development of a DL model that identifies high-risk patients from a clinically comparable 'intermediate-risk' group. ECGs were obtained and archived in an identical manner to cases.

In all SCD cases and controls, paper 12 lead ECG recordings were scanned, and digitized using software (ECGScan), which has been demonstrated to provide a robust reconstruction of a digital ECG waveform[19]. Due to the variable length of ECG leads, we restricted the length of each lead in each sample to a 2.5 s strip, which was the minimum length of ECG waveform for each lead. Hence, digital 2.5 s strips of each lead in the 12 lead ECG were used as input for the DL model. ECGScan produces time series ECG signal, and with a sampling rate of 500 Hz, the final shape of each ECG arrays was $12 \times 1250$.

### Deep learning model development and training
To identify SCD cases using 12 lead ECG waveforms, we developed a convolutional neural network for ECG interpretation (Fig. 2). We trained the model to identify SCD cases with 1,101 prearrest 12 lead ECGs from 1,076 SCD cases from Oregon SUDS and 613 12 lead ECGs from 597 controls. A separate validation cohort of 366 prearrest ECGs and 200 control ECGS was used to determine when to stop model training. The study sample was divided at the patient level so that multiple ECGs from the same patient were included in the same cohort. In the training and validation datasets we used multiple ECGs per patient, but in the internal testing dataset and external validation dataset we used only one ECG per patient (the closest ECG that was unrelated to the SCD event). The mean time from ECG to SCD was $2.0 \pm 2.7$ years in Oregon SUDS and $1.6 \pm 2.1$ years in Ventura PRESTO. We trained the model using the PyTorch DL framework, and the Adam optimizer with default parameters (initial learning rate of 1e-3) with a batch size of 500 and for 55 epochs. Based on the area under the curve of the receiver operating characteristic (AUROC) curve in the validation dataset, we performed early stopping for training.

The DL model was designed to interpret 12 lead ECG waveforms starting with atrous convolutions which were followed by multi-channel 1D convolutions. We limited the number of layers to less than 1/10th the size of previously described architectures[20–23] to minimize model complexity and optimize model runtime. The DL model incorporated convolutional layers after initial atrous layers, with an inverted residual structure. In the DL model, input and output are bottleneck 9 layers with an intermediate expansion layer. To allow information integration across the 12 lead ECGs, the number of input channels increased gradually in each set of expansion layers that were preceded by bottleneck layers. The model was optimized for a lightweight architecture while still maximizing performance[20]. Given that we used ECG waveform instead of images as input data, our deep learning model is a 1D equivalent, and our model is smaller than other ECG models in prior literature with similar performance More details regarding the model architecture can be found in the original papers[20,24].

### Statistical analyses
All continuous variables are expressed as mean ± standard deviation. After model development and training, we performed all statistical analyses on the internal held-out test set and external validation dataset which were never seen during model training. We calculated the model's performance in identifying SCD cases by the AUROC. The model was compared to a previously developed conventional ECG electronic risk score, which evaluates the sum of 6 ECG risk markers: resting heart rate > 75 bpm, LVH, delayed QRS transition, QRS-T angle > 90°, prolonged

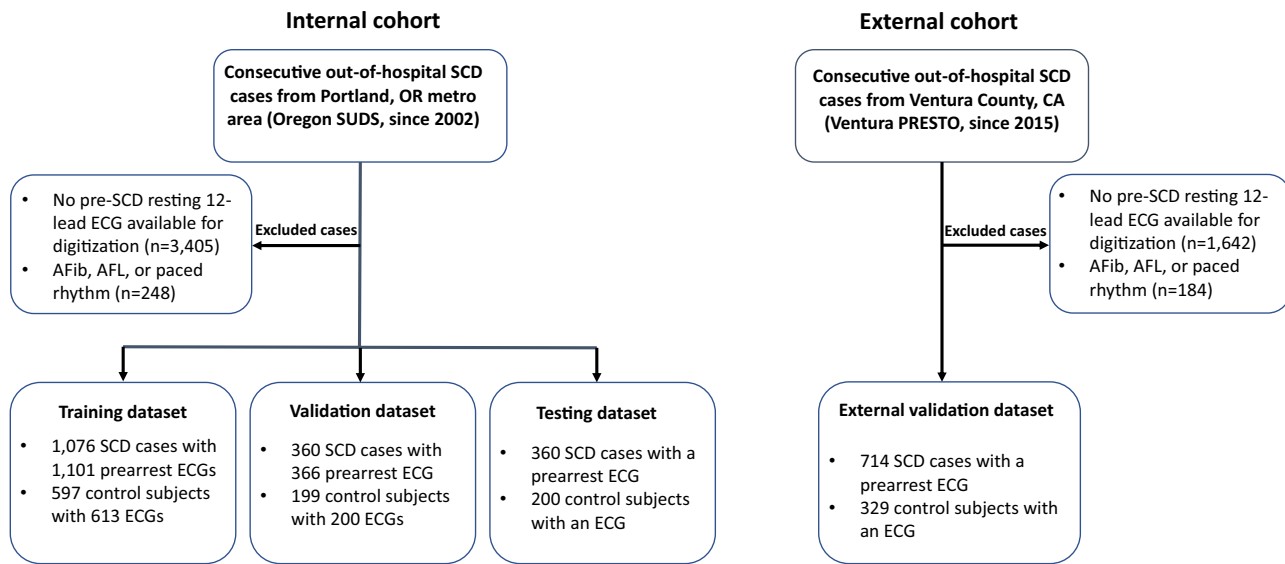

**Fig. 1 | Description of internal and external cohorts.** Study subject selection for the training dataset, validation dataset, testing dataset, and external validation. Afib atrial fibrillation, AFL atrial flutter, ECG electrocardiography, SCD sudden cardiac death.

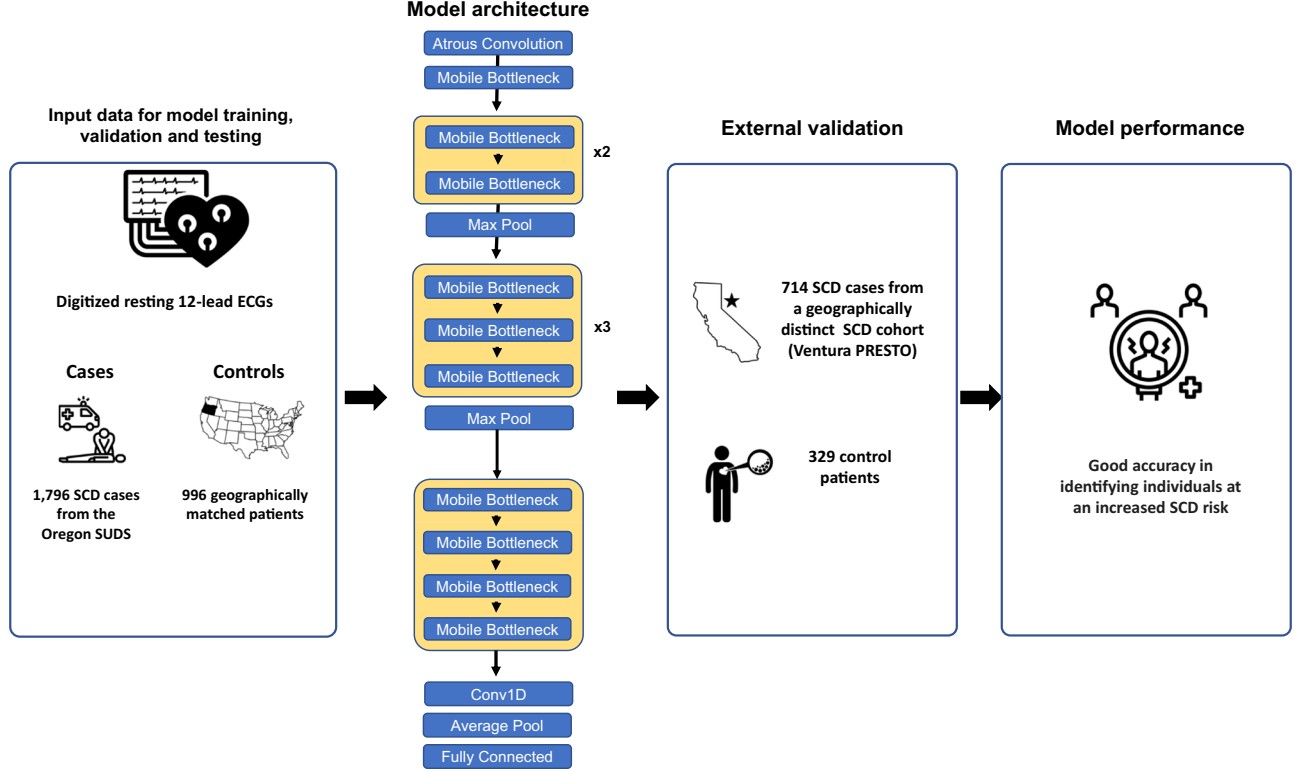

**Fig. 2 | Model development.** Development of deep learning 12-lead ECG model showing input data, model architecture, validation and performance. ECG electro-cardiography, SCD sudden cardiac death.

QTc, and prolonged Tpeak-to-Tend interval[10]. We performed logistic regression models in the internal test dataset and external validation dataset using clinical variables (age, sex, heart failure, coronary artery disease, myocardial infarction, diabetes, chronic obstructive pulmonary disease, seizure, and cerebrovascular accident) with and without DL-ECG analysis output value (DL-ECG index). We selected the best threshold for the model by maximizing the F1 metric on the validation set and used this threshold to report sensitivity and specificity on the test sets. Similarly, the threshold to report sensitivity and specificity for the conventional ECG

electronic risk score and logistic regression models was also selected by maximizing the F1 metric. For each calculation, two-sided 95% confidence intervals (CI) were computed by bootstrapping randomly sampled 50% of the test set for 1,000 iterations. We performed statistical analyses using Python and R.

**Reporting summary**
Further information on research design is available in the Nature Portfolio Reporting Summary linked to this article.

## Results

### Demographic and clinical findings

Our study sample consists of a total of 2,510 SCD cases: 1,796 SCD cases from the Oregon Sudden Unexpected Death Study (SUDS, Portland OR; training, validation, and testing) and 714 SCD cases from the geographically distinct Ventura Prediction of Sudden Death in Multi-ethnic Communities study (PRESTO, Ventura CA; external validation). In comparison to Oregon SUDS SCD cases, Ventura PRESTO SCD cases were older ($72.3 \pm 14.2$ years vs. $67.5 \pm 14.9$ years) and more often female (41.3% vs. 35.4%). The prevalence of Hispanic ethnicity (30.7% vs. 2.4%) and Asian race (7.8% vs. 3.3%) was higher in Ventura PRESTO, while the prevalence of White (82.0% vs. 57.6%) and Black race (10.1% vs. 2.1%) was higher in Oregon SUDS. The prevalence of diabetes was 53.2% in Ventura PRESTO and 45.4% in Oregon SUDS. Previously diagnosed heart failure (31.1% vs. 39.8%) and history of myocardial infarction (MI) (27.5% vs. 38.4%) were lower in Ventura PRESTO compared to Oregon SUDS, respectively. The prevalence of COPD was similar (26.6% in Oregon SUDS vs. 22.3% in Ventura PRESTO).

In comparison to SCD cases, control subjects had a similar prevalence of previously diagnosed coronary artery disease (CAD) (51.2%) and MI (30.7%). However, control subjects were slightly younger ($65.4 \pm 11.6$ years) and had a somewhat lower prevalence of previously diagnosed diabetes (27.8%), atrial fibrillation (13.4%), heart failure (12.8%), and COPD (9.1%). Demographics and clinical characteristics of SCD cases and control subjects are presented in Table 1.

### DL model performance

In the internal testing dataset, the DL model achieved an AUROC of 0.889 (95% CI 0.861–0.917) in detecting SCD cases from controls. Sensitivity and specificity were 0.843 (0.809–0.878) and 0.818 (0.764–0.872), respectively.

In the external validation dataset, the DL model achieved a comparable AUROC of 0.820 (0.794–0.847) in detecting SCD cases. The sensitivity was 0.763 (0.733–0.796), while the specificity was 0.796 (0.753–0.838). Model performance metrics in internal and external cohorts are presented in Table 2 and AUROC curves in Fig. 3.

We evaluated the AUCs of models stratified by sex and age. The DL model performed similarly in men and women in the internal cohort (test for difference in AUCs across subgroups, $p = 0.36$), and marginally better in the external cohort among men (AUC = 0.842, 95% CI 0.81–0.874) than among women (AUC = 0.775, 95% CI 0.718–0.831) ($p = 0.043$). No differences were observed in model performance comparing age > 70 years vs. age ≤ 70 years in the internal or external cohorts ($p = 0.56$ and $p = 0.16$, respectively).

### Conventional ECG electrical risk score performance

We compared the DL model's performance to a previously developed and validated 6-variable ECG electrical risk score that was independently associated with SCD[10]. In the internal and external datasets, the ECG electrical risk score achieved AUROCs of 0.712 (0.668–0.756) and 0.743 (0.711–0.775) in detecting SCD cases from controls, respectively. The sensitivity was 0.779 (0.721–0.837) in the internal testing dataset and 0.569 (0.515–0.623) in the external validation cohort. The specificity was 0.506 (0.454–0.558) in the internal testing dataset and 0.802 (0.773–0.832) in the external validation cohort (Table 2 and Fig. 3).

### Logistic regression models

To evaluate the predictive power of DL-ECG index beyond conventional clinical SCD risk factors, we performed logistic regression analyses including clinical variables with and without DL-ECG index in the internal and external datasets. In the internal test dataset, addition of the DL-ECG index into clinical variables improved the discriminative value of SCD from an AUROC of 0.780 (0.741–0.818) to an AUROC of 0.919 (0.895–0.943). Similar results were obtained in the external test set, in which addition of the DL-ECG index into clinical variables improved the discriminative value of SCD from an AUROC of 0.806 (0.778–0.833) to an AUROC of 0.899 (0.878–0.920). Using a cut-point of 0.70 for predicting case status, the net reclassification improvement using the DL model and clinical variables compared to the model with clinical variables in the internal cohort was 28.7% (95% CI 21.0–36.5%) and in the external cohort was 15.3% (95% CI 9.7–20.9%). Regression model performance metrics in internal and external

**Table 1 | Demographic and clinical characteristics of the study subjects in the internal and external datasets**

| | Internal cohort | | External cohort | |
| --- | --- | --- | --- | --- |
| | SCD cases in Oregon SUDS (*n* = 1796) | Control subjects (*n* = 996) | SCD cases in Ventura PRESTO (*n* = 714) | Control subjects (*n* = 329) |
| Age, years | 67.5 ± 14.9 | 65.3 ± 11.6 | 72.3 ± 14.2 | 65.7 ± 11.7 |
| Female sex, *n* (%) | 635/1796 (35.4%) | 327/996 (32.8%) | 295/714 (41.3%) | 98/329 (29.8%) |
| Race/ethnicity, *n* (%) | | | | |
| White | 1,448/1765 (82.0%) | 836/979 (85.4%) | 411/714 (57.6%) | 273/328 (83.2%) |
| Black | 178/1765 (10.1%) | 105/979 (10.7%) | 15/714 (2.1%) | 33/328 (10.1%) |
| Asian | 59/1765 (3.3%) | 15/979 (1.5%) | 56/714 (7.8%) | 4/328 (1.2%) |
| Hispanic | 43/1765 (2.4%) | 13/979 (1.3%) | 219/714 (30.7%) | 11/328 (3.4%) |
| Other | 37/1765 (2.1%) | 10/979 (1.0%) | 13/714 (1.8%) | 7/328 (2.1%) |
| Prior medical history | | | | |
| Coronary artery disease, *n* (%) | 850/1796 (47.3%) | 502/996 (50.4%) | 257/714 (36.0%) | 176/329 (53.5%) |
| Diabetes, *n* (%) | 815/1795 (45.4%) | 282/984 (28.7%) | 380/714 (53.2%) | 87/328 (26.5%) |
| History of MI, *n* (%) | 689/1796 (38.4%) | 302/996 (30.3%) | 196/714 (27.5%) | 105/329 (31.9%) |
| Atrial fibrillation, *n* (%) | 428/1796 (23.8%) | 137/996 (13.8%) | 187/714 (26.2%) | 41/329 (12.5%) |
| CVA, *n* (%) | 337/1795 (18.8%) | 83/984 (8.4%) | 120/714 (16.8%) | 18/328 (5.5%) |
| Heart failure, *n* (%) | 714/1796 (39.8%) | 136/996 (13.7%) | 222/714 (31.1%) | 33/329 (10.0%) |
| COPD, *n* (%) | 478/1795 (26.6%) | 89/984 (9.0%) | 159/714 (22.3%) | 31/328 (9.5%) |
| Seizure, *n* (%) | 150/1795 (8.4%) | 25/984 (2.5%) | 36/714 (5.0%) | 7/328 (2.1%) |
| Syncope, *n* (%) | 186/1795 (10.4%) | 50/984 (5.1%) | 78/714 (10.9%) | 11/328 (3.4%) |

*COPD* chronic obstructive pulmonary disease, *CVA* cerebrovascular accident, *MI* myocardial infarction, *SCD* sudden cardiac death.

**Table 2 | Model performance in comparison to a conventional ECG electrical risk score in the internal and external datasets**

|  | AUROC (95% CI) | Sensitivity (95% CI) | Specificity (95% CI) | Maximum F1 Metric |
|---|---|---|---|---|
| Internal cohort (*n* = 2,792) |  |  |  |  |
| Deep learning model | 0.889 (0.861–0.917) | 0.843 (0.809–0.878) | 0.818 (0.764–0.872) | 0.866 |
| Conventional ECG electrical risk score | 0.712 (0.668–0.756) | 0.779 (0.721–0.837) | 0.506 (0.454–0.558) | 0.585 |
| External cohort (*n* = 1,043) |  |  |  |  |
| Deep learning model | 0.820 (0.794–0.847) | 0.763 (0.733–0.796) | 0.796 (0.753–0.838) | 0.823 |
| Conventional ECG electrical risk score | 0.743 (0.711–0.775) | 0.569 (0.515–0.623) | 0.802 (0.773–0.832) | 0.571 |

*AUROC area under the receiver operating characteristics curve, CI Confidence interval.*

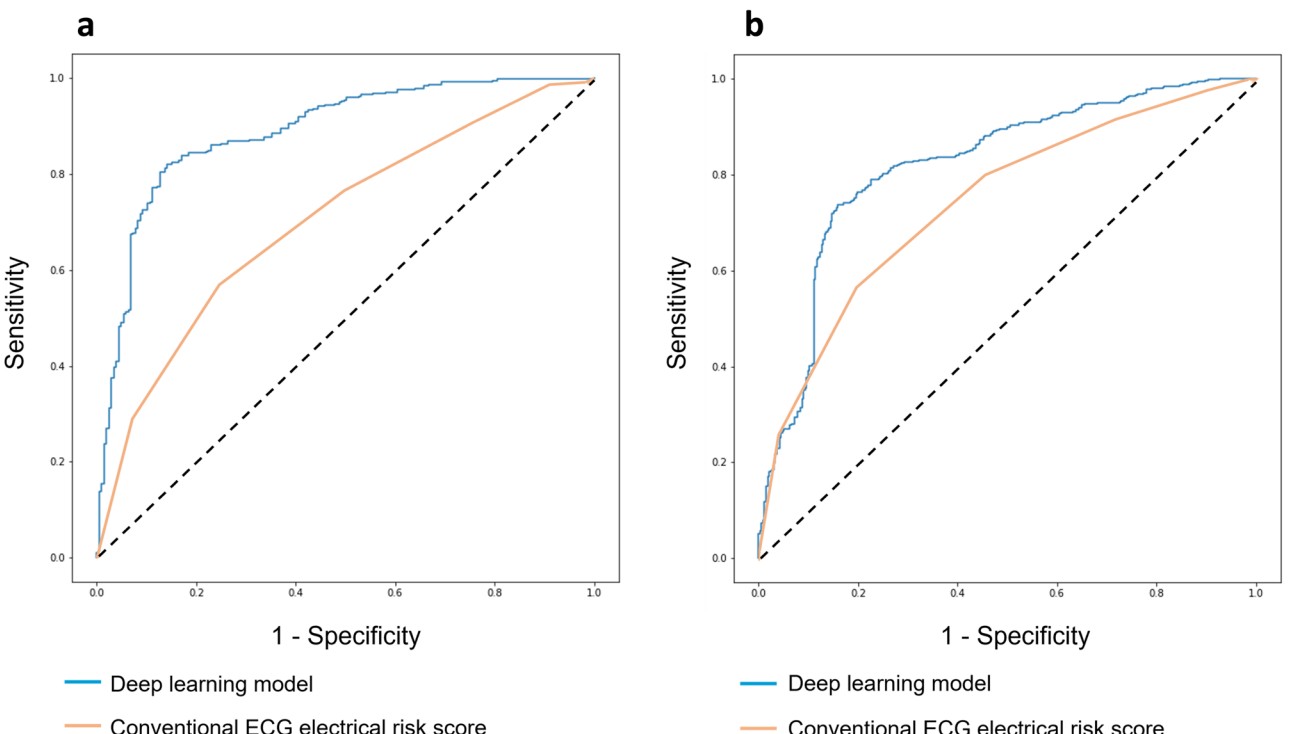

**Fig. 3 | Comparison of deep learning model with conventional risk score.** The deep learning ECG model was compared with a conventional ECG risk score for prediction of sudden cardiac death (SCD). Receiver operating curves for the identification of SCD cases in the internal (**a**, *n* = 2792) and external cohort (**b**, *n* = 1043).

cohorts are presented in Table 3 and AUROC curves in Fig. 4. Examples of Local Interpretable Model-agnostic Explanations (LIME) highlighted ECGs from two SCD cases and two controls from the external cohort are presented in the Supplementary Figure. LIME highlighted a wide range of ECG features including PR interval, QRS complex, and ST- and T-wave changes.

## Discussion

We utilized data from two large geographically distinct community-based out-of-hospital SCD cohorts to train, test, and validate a 12 lead ECG waveform-based DL model, which was compared to a previously validated conventional ECG model. The DL model achieved a higher accuracy with an AUROC of 0.889 for internal cohort and 0.820 for external validation, and outperformed the conventional ECG risk score. A slightly lower performance in the external cohort may be related to differences in demographics and clinical profiles, which may affect the accuracy of the model's prediction. However, despite such differences, the overall performance remained good in the external cohort. To our knowledge, this is the first report of an ECG-based DL model that has outperformed a conventional ECG risk model in predicting out-of-hospital SCD at the community level.

There are some unique aspects of study design that made this work feasible. SCD is a dynamic and unexpected event that requires prospective

ascertainment[17]. Since annual incidence is in the range of 50–100/100,000[1], existing cohorts of 5000–10,000 subjects cannot yield sufficient numbers of SCD cases for viable analyses, especially those that employ deep learning models. Furthermore, we were able to include both survivors and non-survivors of SCD in our datasets, which avoids the bias of predicting only non-survivors or survivors. The establishment of the two population cohorts Oregon SUDS[17] and Ventura PRESTO[18] consisting of ~1.85 million US residents, provided sufficient numbers for deep learning. Equally important, both studies have been obtaining and archiving digitized 12 lead ECGs performed prior, and unrelated to SCD events. While this is a challenging process for the SCD phenotype, it is a pre-requisite for discovery of prediction models.

Recently published studies have developed DL models to predict SCD or ventricular arrhythmias (VA) using cardiac MRI images[25,26], monophasic action potentials recorded during invasive electrophysiology studies[27] as well as clinical data[28], that also show promise. The major advantages of utilizing the resting 12 lead ECG for SCD risk assessment relate to low cost, noninvasiveness, and wide availability in diverse populations around the globe. Given the low cost and wide availability of ECGs, this model has potential to augment early screening for subclinical cardiovascular conditions that carry an increased SCD risk. More recently, rapid developments in

**Table 3 | Performance of logistic regression models including clinical variables with and without deep learning ECG index in the internal and external datasets**

| | AUROC (95% CI) | Sensitivity (95% CI) | Specificity (95% CI) | Maximum F1 Metric |
|---|---|---|---|---|
| **Internal cohort (n = 2,792)** | | | | |
| Clinical variables | 0.780 (0.741–0.818) | 0.773 (0.714–0.831) | 0.644 (0.595–0.694) | 0.639 |
| Clinical variables + Deep learning ECG index | 0.919 (0.895–0.943) | 0.763 (0.703–0.822) | 0.914 (0.885–0.943) | 0.794 |
| **External cohort (n = 1,043)** | | | | |
| Clinical variables | 0.806 (0.778–0.833) | 0.777 (0.732–0.823) | 0.702 (0.668–0.735) | 0.641 |
| Clinical variables + Deep learning ECG index | 0.899 (0.878–0.920) | 0.808 (0.765–0.851) | 0.842 (0.815–0.869) | 0.751 |

Clinical variables include age, sex, heart failure, coronary artery disease, myocardial infarction, diabetes, chronic obstructive pulmonary disease, seizure, and cerebrovascular accident. *AUROC* area under the receiver operating characteristics curve, *CI* Confidence interval.

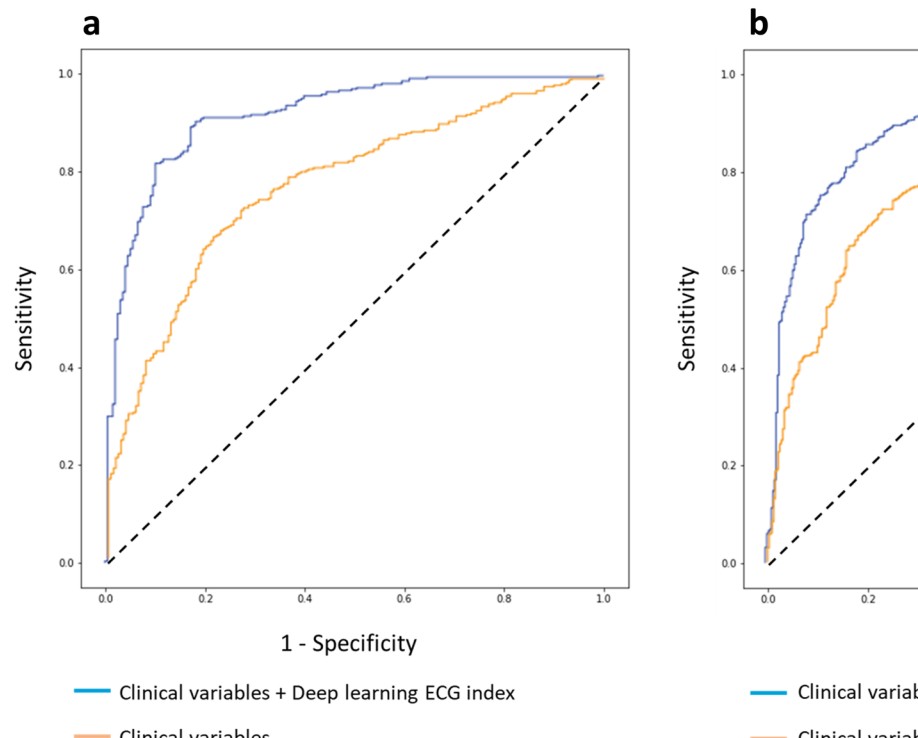
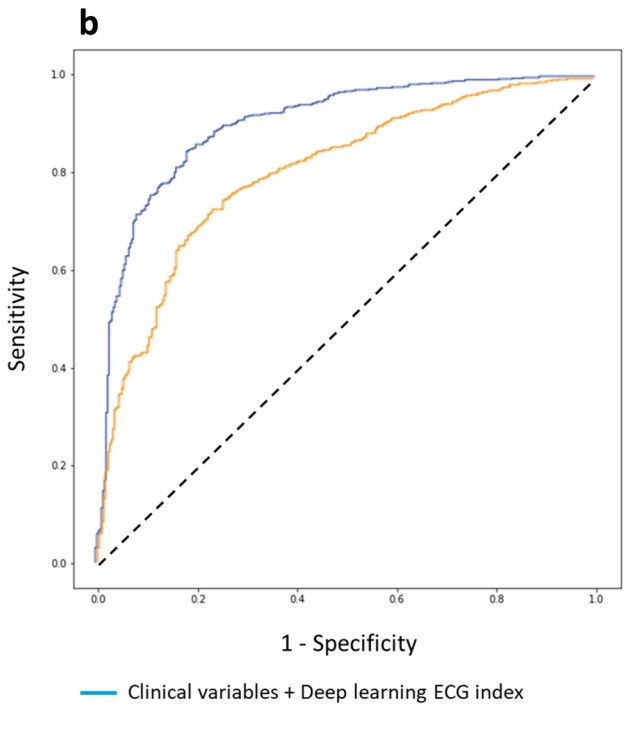

**Fig. 4 | Utility of deep learning ECG index beyond clinical risk predictors.** Receiver operating curves for the identification of sudden cardiac death cases in the internal (**a**, n = 2792) and external cohort (**b**, n = 1043) with logistic regression models. Clinical variables include age, sex, heart failure, coronary artery disease, myocardial infarction, diabetes, chronic obstructive pulmonary disease, seizure, and cerebrovascular accident.

wearable device technology have enabled recording of the ECG beyond the healthcare environment, during activities of daily living.

As compared to a previously developed and validated 6 variable conventional ECG risk score, the DL model achieved significantly higher performance in detecting SCD cases, which supports the higher utility of DL-based models. In contrast to conventional risk calculators, DL models do not require manual feature selection and extraction but instead can utilize the entire digital signal to incorporate novel indices of risk. Consequently, ECG DL models are not biased in focusing on pre-specified ECG parameters and thus have the potential to achieve higher throughput and broader scope while preserving accuracy (Fig. 5). Another major advantage of DL techniques in comparison to conventional statistical tools is that they require making fewer assumptions about data structure. Hence, DL models can be more accurate for evaluation of complex nonlinear relationships in large datasets. However, DL techniques may also have some disadvantages which need to be considered during development, and also prior to deployment. These include model-specific requirements for inputting data, vulnerability to

systemic bias and lack of the ability to explain mechanisms of findings which is still a work in progress.

SCD is a complex trait as well as a multifactorial event, and pathophysiology is based on the interplay between the underlying substrate and a variety of triggers. ECG abnormalities that have been associated with an increased risk of SCD are often surrogates of the underlying cardiac substrate (e.g., LVH, myocardial scarring, repolarization abnormality), and accurate risk stratification requires a combination of several nonspecific ECG abnormalities[10,11]. Even though ECG may reflect widespread cardiac and noncardiac conditions[29–31], the logistic regression model showed that DL-ECG index improved the discriminative value of SCD over clinical variables. Similar findings were found in a recent DL-ECG model among heart failure patients[32]. In comparison to conventional dichotomous analytical methods, deep learning-based ECG analysis may provide more precise and comprehensive quantification of ECG abnormalities and deeper phenotyping.

The vast majority of SCD cases are not identified as high risk prior to their mostly lethal event, which highlights the importance of extending SCD risk assessment beyond left ventricle ejection fraction. While the overall

**Fig. 5 | Summary of deep learning versus conventional ECG analysis.** Schematic illustration of the advantage of deep learning-based ECG analysis over conventional methods. LVH left ventricular hypertrophy, SCD sudden cardiac death.

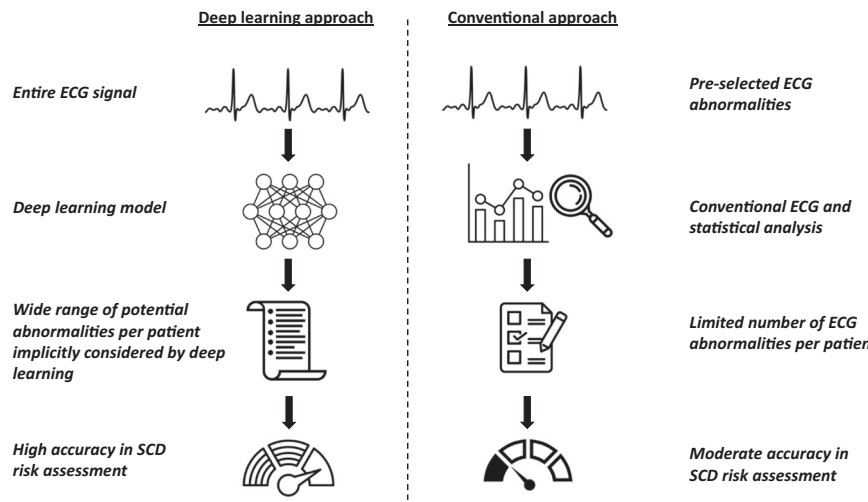

incidence of SCD in the community has remained relatively stable, the incidence of SCD in heart failure with reduced ejection fraction has decreased[6], suggesting that the use of LVEF < 35% in risk prediction is progressively less effective. An effective risk stratification would require both optimal screening population and accurate screening tools, which is likely to consist of a combination of several risk assessment modalities (e.g., ECG, imaging, omics, etc.). ECG abnormalities have already been included in recent SCD risk models[8], and usage of DL based ECG analysis as a pre-screening tool for identification of individuals who could be triaged for more comprehensive risk evaluation could prove effective in selected individuals. Due to the high proportion of CAD and prior MI without a history of ventricular arrhythmias, we think that the control subjects in the present study represent intermediate-risk patients, and given the increased baseline SCD risk, this patient group may represent a reasonable target for screening efforts. However, our study represents the first steps in ECG-DL based SCD risk assessment. As for all DL models that have the potential to be clinically useful, further prospective studies followed by randomized trials are needed to study if DL-based ECG analysis has the potential to provide an inexpensive, high throughput, and widely deployable pre-screening tool to augment current SCD risk stratification.

A strength of our study is a large sample of carefully adjudicated out-of-hospital SCD cases with prearrest resting 12 lead ECG. In addition, we were able to create balanced datasets by including clinically comparable control patients. However, some limitations should be considered while interpreting these findings. We matched cases and controls geographically in the training, validation, and internal testing datasets, but we had no geographically matched controls in external validation dataset. However, there was no overlap of control samples between the internal and external validation datasets. The necessity of using a large control group with digitized paper ECGs divided into two was driven by the goal of minimizing the differences in the quality of ECG recordings between cases and controls. The Oregon SUDS study was initiated in 2002 when only paper ECGs were made available, and all case and control ECGs were digitized before providing them to the DL-ECG model. Control ECGs were also obtained randomly from multiple community hospitals and health systems, which further reduces the likelihood of systemic bias in quality of ECG recordings. Since SCD is the first manifestation of heart disease in a substantial subgroup, a prearrest ECG was not available if individuals had not undergone a cardiac evaluation prior to their SCD event, creating potential for selection bias. Although we aimed to match cases and controls based on the underlying CAD status, some differences in other SCD risk factors remained between SCD cases and controls, which may have affected the model performance. However, some of these differences are important contributors to the development of SCD, and the prediction of SCD is based on the identification and combination of risk markers. We used a case-control study design to collect sufficient numbers of carefully adjudicated SCD cases, which does not allow us to reliably estimate the positive predictive value and negative predictive value. We used a relatively short 2.5 s ECG strip, and further studies are probably needed to investigate whether longer ECG strip usage will result in higher prediction accuracies. Our model is only applicable to sinus rhythm ECGs since atrial fibrillation/flutter and paced ECGs were excluded during algorithm development. Lastly, the majority of our study subjects were White, and further studies are needed to study ECG-DL performance in racially/ethnically distinct subgroups. Additionally, future prospective studies are needed to validate model performance in clinically diverse settings.

## Conclusions

We trained an ECG-based DL model that achieved high accuracy in distinguishing SCD cases from control patients. The model was successfully validated in a geographically distinct SCD cohort and outperformed a previously validated conventional ECG risk score. These results suggest that DL-based ECG analysis has advantages over conventional ECG based SCD risk assessment and yields better accuracy. Further detailed investigation is warranted to evaluate how the DL model could contribute to improved SCD risk stratification.

## Data availability

All analytical methods applied for the deep learning algorithm are included in this published article. Based on institutional review board guidance patient data is not publicly available and is de-identified. De-identified data is only available by contacting the corresponding author.

## Code availability

Code is available at https://github.com/ecg-net/scd-oregon[33]

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

## Acknowledgements

This work is funded, in part, the by National Institutes of Health, National Heart Lung and Blood Institute Grants R01HL145675 and R01HL147358 to SSC. SSC holds the Pauline and Harold Price Chair in Cardiac Electrophysiology at Cedars-Sinai. LH is a postdoctoral fellow visiting from the Research Unit of Internal Medicine, Medical Research Center Oulu, University of Oulu and Oulu University Hospital, Oulu, Finland, and is funded by the Sigrid Juselius Foundation, the Finnish Cultural Foundation, the Instrumentarium Science Foundation, the Orion Research Foundation, and the Paavo Nurmi Foundation. The funding sources had no involvement in the preparation of this work or the decision to submit it for publication.

## Author contributions

Study design and conception: L.H., D.O., and S.S.C. Acquisition, analysis or interpretation of data: L.H., H.C., K.N., Z.B., M.S., A.U-E., K.R., D.O., and S.S.C. Drafting of the manuscript: L.H., and S.S.C. Statistical analysis: H.C., K.R., D.O., and S.S.C. Critical revision of the manuscript for important intellectual content: L.H., H.C., K.N., Z.B., M.S., A.U-E., K.R., D.O., and S.S.C. Administrative and material support: D.O., and S.S.C. Obtained funding: S.S.C. Supervision: S.S.C. Full access to the data: S.S.C. All authors reviewed and approved the final version of the manuscript.

## Competing interests

The authors declare the following competing interests: The deep learning techniques for predicting sudden cardiac death risk discussed in this manuscript relate to pending US provisional patent application 63/500,550 naming Cedars-Sinai Medical Center as the applicant and listing S.S.C and D.O. as the inventors. The remaining authors declare no competing interests.
