## [Peer Review File · Communications Medicine]

Reviewers' comments:

Reviewer #1 (Remarks to the Author):

This manuscript by Holmstrom et al. describes the use of machine learning via deep convolutional neural networks (DCNN) to predict risk of sudden cardiac death (SCD) from 12-lead ECG. The authors followed many aspects of best practices in machine learning by using two different test sets (internal, external) from large SCD cohorts and compared those to control datasets from other large cohorts including at least 50% with significant CAD. DCNN model performance is evaluated via ROC analysis, which suggested the new approach was superior to a conventional ECG models.

Big-picture Concerns:

1. More information about the use of ECGScan would be helpful. Did this produce digital image files (i.e., high resolution images) from paper ECG traces? Or were the outputs of the process time series data? Were all 12 leads of the ECG digitized? Only eight of those signals are linearly independent so including all 12 will result in redundant information being scrutinized by the DCNN (i.e., one of the limb leads and all of the augmented limb leads can be reconstructed from linear combinations of the other signals).

2. Further to the above point, explicit clarifying information should be provided on what exactly was ingested by the DCNN. Was it one image file per patient ECG or was it 8/12 time series (representing different leads)? If image files were used as input, was there a need to remove annotations so that the DCNN would consider exclusively the ECG information itself and not inadvertently features present in the annotations? Examples (in the supplemental material?) showing exactly what is ingested by the DCNN could be a straightforward way to address this concern.

3. Lines 144-147 and second panel of Fig. 2: Overall, the factors supporting the rationale for model architecture are not discussed. More information is needed to ensure that a reasonably tech-savvy reader could understand what is being done, as well as why, and attempt to reimplement or extend the work themselves. The explanations of "bottleneck layers," "inverted residual structure," and "atrous convolutions" are insufficient for reproducibility. The same is true for the overall description of and rationale for model architecture. Why did the authors opt to insert max pool layers after the blocks of mobile bottleneck layers? Does the labeling mean there were four mobile bottleneck blocks in the first yellow box and nine mobile bottleneck blocks in the second yellow box (i.e., 2x2 and 3x3)? If not, the "x2" and "x3" labels are likely redundant. How did the authors select the initial learning rate (1e-3), the batch size (500), and the number of training epochs (55)?

Minor Concerns:

4. Line 158, "DL-ECG" index has never been properly defined.

5. Lines 185-192, Lines 201-210, associated tables, and elsewhere: AUROC is a reasonable metric, especially for datasets that are relatively well-balanced (as is the case in this study). Nevertheless, the authors should strongly consider showing other metrics for robustly evaluating model performance here, including F1-score, PPV, and NPV. F1-score is of particular interest since the authors state that was the performance metric used to select optimal model threshold. In addition to text in the Results/Tables sections, F1-score should be reported alongside AUROC in other

prominent parts of the paper (e.g., Abstract, Discussion).

6. Lines 216-217: "To our knowledge, this is the first report of an ECG-based DL model that has outperformed a conventional ECG risk model." Other examples exist. For instance, Sridhar et al. (2022) *Cardiovasc Digit Health J* compared DCNN vs. conventional model performance for ECG-based prediction of adverse outcomes in COVID-19. This does not preempt the importance of the work carried out by the authors in the present study, but caution should be exercised when making statements of primacy and/or priority.

7. Fig. 5: The schematic suggests that through DL-based analysis there is a "wide range of abnormalities identified per patient". Although likely true, this statement is not directly supported by the data presented in the manuscript. Which features are being "identified"? This suggests a higher level of explainability and interpretability than what is possible with DCNN tools. It would be more accurate to say "wide range of potential abnormalities in individual patients implicitly considered by ML," or something like that.

Reviewer #2 (Remarks to the Author):

I appreciate the opportunity to provide feedback on this manuscript, which offers a comprehensive analysis of a deep learning model aimed at predicting SCD risk from ECG data. The study's strengths lie in its diverse dataset, sound statistical analyses, and meaningful comparisons with conventional ECG risk scores.

One of the paper's key strengths is the in-depth discussion of the results coupled with a clear outline of the study's strengths and limitations. The thoughtful selection of the control population, ensuring a similar prevalence of coronary artery disease in the control and SCD groups, is sound.

Certain areas would benefit from further elaboration:

1. It would be useful to have a more detailed explanation for the exclusion of ECGs with atrial fibrillation or atrial flutter and how alternative ECGs were chosen.
2. Figure 1 doesn't show how many cases were excluded, if reasonable, consider inserting this.
3. Discussion on why the model performances varied between internal and external validation cohorts and whether the authors had any insights to why this occurred, would be helpful.
4. I recommend a comment on whether the authors felt using a 2.5-second ECG strip was a limitation, would be helpful.
5. Line 170-178: It's unclear whether the age, sex, and health condition differences between the two geographically distinct populations were taken into account in the model. This should be commented on in the limitation section if not.
6. For a clearer understanding of the mechanism, further discussion of how DL models can utilise the entire digital signal vs digital scans of ECGs as in this study, would be a helpful addition in the discussion section.
7. Further exploration of how the model could be applied in different clinical settings and its potential effect on patient outcomes would amplify the paper's relevance.
8. While the overall performance of the DL model is well-presented, discussing specific categories where the model performed exceptionally well or poorly could provide insight into its strengths and

weaknesses.

9. Line 280-285: As limitations, the authors have mentioned some biases that might have affected the results. It would be helpful if they would propose some ways to reduce these biases in future studies.

The authors should be praised for a thorough analysis of the results and potential implications for SCD prediction. The authors have demonstrated a study that is a valuable addition to the cardiology field and I recommend publication.

Reviewer #3 (Remarks to the Author):

The authors set out to develop a deep learning (DL) model on 12-lead ECGs to stratify for SCD risk compared to conventional ECG risk models. From the SUDS study in Portland Oregon, they identified 1827 out-of-hospital SCD ECGs from 1,796 cases. They validated the model in a separate cohort in Ventura County, CA, in 714 ECGs from 714 SCD cases. Two separate control group samples were obtained from 1,342 ECGs taken from 1,325 individuals of whom at least 50% had significant coronary disease. The DL model was compared with a previously validated conventional 6-variable ECG risk model. The DL model achieved an AUROC of 0.889 (95% CI 0.861-0.917) for the detection of SCD cases vs. controls in the internal hold-out dataset, and was successfully validated in external SCD cases with an AUROC of 0.820 (0.794-0.847). The DL model performed significantly better than the conventional ECG model that achieved an AUROC of 0.712 (0.668- 0.756) in the internal and 0.743 (0.711-0.775) in the external cohort. They conclude that an ECG-based DL model distinguished SCD cases from controls with improved accuracy compared to a conventional ECG risk model.

This is a nice study.

- In the abstract and manuscript, it is incorrect to state “prospectively identified” since the analysis was run on already-collected data.
- Were all cases from SUDS and Ventura included – which if any were excluded?
- Were control cases propensity matched – on which variables?
- This is not clear – since training is usually completed prior to validation: “Based on the area under the curve of the 138 receiver operating characteristic (AUROC) curve in the validation dataset, we performed early 139 stopping for training”. Please clarify.
- Please provide details of model dropout and architecture. “We limited the number of layers to less than 1/10th 142 the size of previously described architectures(20-22) to minimize 143 model complexity and optimize model runtime.” Please provide code as a supplement.
- Please report predictive accuracy at different points in time.
- Few ECG models are used to guide risk stratification beyond LVEF. It is not clear that a model that improves upon ECG index alone adds to models including LVEF. Is the ECG comparator useful?
- Table 3. Please construct a model including SCD risk factors of LVEF and existing ECG risk factors e.g. QRS duration, history of AF – did DL add to predictions of such a model?
- What is net reclassification rate using DL model compared to the clinically useful model above?
- Please report explainability of trained DL model – at least which regions of the ECG by Shapley or other metrics that most contributed to predictive value.

Reviewers' comments:

Reviewer #1 (Remarks to the Author):

This manuscript by Holmstrom et al. describes the use of machine learning via deep convolutional neural networks (DCNN) to predict risk of sudden cardiac death (SCD) from 12-lead ECG. The authors followed many aspects of best practices in machine learning by using two different test sets (internal, external) from large SCD cohorts and compared those to control datasets from other large cohorts including at least 50% with significant CAD. DCNN model performance is evaluated via ROC analysis, which suggested the new approach was superior to a conventional ECG models.

Big-picture Concerns:

1. More information about the use of ECGScan would be helpful. Did this produce digital image files (i.e., high resolution images) from paper ECG traces? Or were the outputs of the process time series data? Were all 12 leads of the ECG digitized? Only eight of those signals are linearly independent so including all 12 will result in redundant information being scrutinized by the DCNN (i.e., one of the limb leads and all of the augmented limb leads can be reconstructed from linear combinations of the other signals).

Reply: Thank you for these excellent comments. ECGScan is an established method used in multiple prior publications over 17 years (PMID 16216602) which produces a high resolution, time series digital ECG signal. All 12 leads were digitized and given that we had a 2.5-second strip for each lead with a sampling rate of 500Hz, the final shape of the ECG arrays was 12 x 1250. Our decision to include all 12 leads is based on the standard ECG recording in clinical practice, and previous studies on DL-based ECG evaluation of cardiac disorders have used all 12 leads as well. We have now stated this more clearly in the Methods section (Page 6, Line 6).

2. Further to the above point, explicit clarifying information should be provided on what exactly was ingested by the DCNN. Was it one image file per patient ECG or was it 8/12 time series (representing different leads)? If image files were used as input, was there are need to remove annotations so that the DCNN would consider exclusively the ECG information itself and not inadvertently features present in the annotations? Examples (in the supplemental material?) showing exactly what is ingested by the DCNN could be a straightforward way to address this concern.

Reply: Excellent question. We provided 2.5s time series data for 12 ECG leads with a sampling rate of 500Hz. We did not use image files. The final shape of each ECG array was 12 x 1250. We have now clarified this in the Methods (Page 6, Line 6).

3. Lines 144-147 and second panel of Fig. 2: Overall, the factors supporting the rationale for model architecture are not discussed. More information is needed to ensure that a reasonably tech-savvy reader could understand what is being done, as well as why, and attempt to reimplement or extend the work themselves. The explanations of “bottleneck layers,” “inverted residual structure,” and “atrous convolutions” are insufficient for reproducibility. The same is true for the overall description of and rationale for model architecture. Why did the authors opt to insert max pool layers after the blocks of mobile bottleneck layers? Does the labeling mean there were four mobile bottleneck blocks in the first yellow box and nine mobile bottleneck blocks in the second yellow box (i.e., 2x2 and 3x3)? If not, the “x2” and “x3” labels are likely redundant. How did the authors select the initial learning rate (1e-3), the batch size (500), and the number of training epochs (55)?

Reply: Thank you for this comment. We designed the model architecture based on sweep of architecture size vs. performance from the EfficientNet paper (arXiv:1905.11946v5) and optimized for a light weight architecture while still maximizing performance. This is a 1D (given waveform instead of images) equivalent that is smaller than other ECG models with similar performance. More details regarding the model architecture can be found in our original paper (arXiv:2205.03242).

We have now provided the code and a more detailed rationale for the model architecture in the Supplemental material.

Minor Concerns:

4. Line 158, “DL-ECG” index has never been properly defined.

Reply: Good catch. We have now defined this in the Methods section (Page 7, Line 17).

5. Lines 185-192, Lines 201-210, associated tables, and elsewhere: AUROC is a reasonable metric, especially for datasets that are relatively well-balanced (as is the case in this study). Nevertheless, the authors should strongly consider showing other metrics for robustly evaluating model performance here, including F1-score, PPV, and NPV. F1-score is of particular interest since the authors state that was the performance metric used to select optimal model threshold. In addition to text in the Results/Tables sections, F1-score should be reported alongside AUROC in other prominent parts of the paper (e.g., Abstract, Discussion).

Reply: Thank you for this comment. We have now provided F1 scores for each model alongside the AUROC values in Table 2 and Table 3. However, as you know F1 score, PPV and NPV may not be a straightforward calculation in case-control studies. Disease prevalence is a required component of

the calculation, and by design, a case-control study does not typically reflect the disease prevalence in the target population. Hence, F1-score, PPV, and NPV values may not be directly generalizable to a population setting, since the proportion of SCD cases in the overall population is low. Using an established population-based case-control approach, we were able to collect a high number of out-of-hospital SCD cases, which was an important part of our goal to avoid imbalanced datasets and create a model that is primarily able to detect SCD cases instead of non-SCD cases. In other words, a more 'realistic' prevalence with >90% of controls could bias a model toward the detection of controls instead of cases at high SCD risk.

6. Lines 216-217: "To our knowledge, this is the first report of an ECG-based DL model that has outperformed a conventional ECG risk model." Other examples exist. For instance, Sridhar et al. (2022) Cardiovasc Digit Health J compared DCNN vs. conventional model performance for ECG-based prediction of adverse outcomes in COVID-19. This does not preempt the importance of the work carried out by the authors in the present study, but caution should be exercised when making statements of primacy and/or priority.

Reply: Good point. It would be more accurate to state that this is the first report of an ECG-based DL model that has outperformed a conventional model in predicting out-of-hospital SCD at the community level. We have now revised our statement in the Discussion (Page 10, Line 21).

7. Fig. 5: The schematic suggests that through DL-based analysis there is a "wide range of abnormalities identified per patient". Although likely true, this statement is not directly supported by the data presented in the manuscript. Which features are being "identified"? This suggests a higher level of explainability and interpretability than what is possible with DCNN tools. It would be more accurate to say "wide range of potential abnormalities in individual patients implicitly considered by ML," or something like that.

Reply: An excellent point. We have now revised Figure 5 as suggested and revised "wide range of abnormalities identified per patient" to "wide range of potential abnormalities per patient implicitly considered by deep learning".

Reviewer #2 (Remarks to the Author):

I appreciate the opportunity to provide feedback on this manuscript, which offers a comprehensive analysis of a deep learning model aimed at predicting SCD risk from ECG data. The study's strengths lie in its diverse dataset, sound statistical analyses, and meaningful comparisons with conventional ECG risk scores.

One of the paper's key strengths is the in-depth discussion of the results coupled with a clear outline of the study's strengths and limitations. The thoughtful selection of the control population, ensuring a similar prevalence of coronary artery disease in the control and SCD groups, is sound.

Certain areas would benefit from further elaboration:

1. It would be useful to have a more detailed explanation for the exclusion of ECGs with atrial fibrillation or atrial flutter and how alternative ECGs were chosen.

Reply: Thank you for your excellent comments. ECGs with paced rhythm, atrial fibrillation, or atrial flutter were excluded a priori to create a DL model that could be applied to ECGs in sinus rhythm. Including those ECGs may lead to a biased DL model that could learn to recognize paced rhythm, atrial fibrillation, or atrial flutter, instead of an increased SCD risk. Our goal was to create a DL model that would identify a high SCD risk based on the ECG morphology, and hence, excluding paced rhythm, atrial fibrillation, and atrial flutter made the groups more comparable. We have now clarified this in the Methods section (Page 5, Line 8).

2. Figure 1 doesn't show how many cases were excluded, if reasonable, consider inserting this.

Reply: Good point. We have now added the numbers of excluded cases (no pre-SCD ECG available for digitization or ECG with atrial fibrillation, atrial flutter, or paced rhythm) in Figure 1.

3. Discussion on why the model performances varied between internal and external validation cohorts and whether the authors had any insights to why this occurred, would be helpful.

Reply: Great comment. The performance of AI models is generally lower in external validation cohorts compared to the internal cohort. In our study, a likely reason is that patient demographics and clinical profiles differed significantly between the internal and external cohorts. For example, higher mean age, a higher proportion of females and Hispanic ethnicity, and a lower prevalence of prior MI and heart failure in the external SCD cohort may have had a modest effect on the accuracy of the model's predictions. However, despite the demographic and clinical differences between the internal and external cohorts, our model was successfully validated which indicates good generalizability and mitigates the possibility of overfitting or systematic biases. We have now addressed this in more detail in the Discussion (Page 10, Line 17).

4. I recommend a comment on whether the authors felt using a 2.5-second ECG strip was a limitation, would be helpful.

Reply: Excellent suggestion. While a 2.5-second ECG can be considered relatively short, from our perspective it is enough to capture a representative sample of the ECG morphology and develop a model to predict SCD. Our results also suggest that a 2.5-second ECG strip may provide enough information for a DL-based ECG risk assessment. However, this is a good point, and further studies are probably needed to investigate whether a longer ECG strip results in higher model accuracies. We have now commented on this in the limitations (Page 14, Line 8).

5. Line 170-178: It's unclear whether the age, sex, and health condition differences between the two geographically distinct populations were taken into account in the model. This should be commented on in the limitation section if not.

Reply: Good point. The external SCD cohort was intentionally recruited from a distinct geographical site with different demographics to assess the generalizability of the internal SCD cohort findings. We would submit that successful external validation despite the age, sex, and health condition differences between the two geographically distinct populations denotes good model generalizability. Hence, the differences in subject profiles between internal and external SCD cohorts can be considered a strength rather than a limitation. We have addressed this in the Discussion (Page 10, Line 17).

6. For a clearer understanding of the mechanism, further discussion of how DL models can utilise the entire digital signal vs digital scans of ECGs as in this study, would be a helpful addition in the discussion section.

Reply: Thank you for this comment. Conventional ECG risk scores usually require dichotomous pre-specified ECG variables, whereas ECG DL models are provided with the entire ECG signal. In other words, DL models do not require manual feature selection and extraction but instead can utilize the entire digital signal to incorporate potentially novel indices of risk. Hence, ECG DL models are not biased in focusing on pre-specified ECG parameters with the potential to achieve higher throughput and broader scope while preserving accuracy. We have addressed this in the Discussion (Page 11, Line 19).

7. Further exploration of how the model could be applied in different clinical settings and its potential effect on patient outcomes would amplify the paper's relevance.

Reply: An excellent comment. Over time, the current left ventricular ejection fraction-based SCD risk stratification approach has become insufficient for identifying patients at high risk of sudden cardiac death who are most likely to benefit from primary prevention. DL-based ECG analysis has potential for use as a pre-screening tool for the identification of individuals who could be triaged for more comprehensive risk evaluation. However, further prospective studies followed by randomized trials are needed to investigate if DL-based ECG analysis has the potential to provide an inexpensive, high

throughput, and widely deployable pre-screening tool to augment current SCD risk stratification. We have now included this aspect in the Discussion (Page 12, Line 18).

8. While the overall performance of the DL model is well-presented, discussing specific categories where the model performed exceptionally well or poorly could provide insight into its strengths and weaknesses.

Reply: An excellent point. Based on this suggestion, we evaluated the performance of the DL model in different patient subgroups stratified by sex and age. In the internal cohort, AUCs for the internal cohort were similar among women (AUC=0.903, 95% CI 0.857-0.949) and men (AUC=0.875, 95% CI 0.837-0.913) ($p=0.36$) and by age group: among those age >70 years (AUC=0.872, 95% CI 0.821-0.924) and age ≤ 70 (AUC=0.891, 95% CI 0.855-0.927) ($p=0.56$). In the external cohort, the DL model performed somewhat better among men (AUC=0.842, 95% CI 0.81-0.874) than among women (AUC=0.775, 95% CI 0.718-0.831) ($p=0.043$) but did not differ by age group: age >70 years (AUC=0.800, 95% CI 0.753-0.847) and age ≤ 70 (AUC=0.842, 95% CI 0.806-0.878) ($p=0.16$). We have now included these findings in the Results section (Page 9, Line 5).

9. Line 280-285: As limitations, the authors have mentioned some biases that might have affected the results. it would be helpful if they would propose some ways to reduce these biases in future studies.

Reply: Thank you for this comment. We used a case-control design to develop a DL model and we propose that future investigations should focus on validating such models in prospective studies. Also, the model performance should be studied in other racially/ethnically diverse groups. We have discussed this at the end of the Limitations (Page 14, Line 12).

The authors should be praised for a thorough analysis of the results and potential implications for SCD prediction. The authors have demonstrated a study that is a valuable addition to the cardiology field and I recommend publication.

Reply: Thank you for this comment. We appreciate your constructive criticism and careful review of our work.

Reviewer #3 (Remarks to the Author):

The authors set out to develop a deep learning (DL) model on 12-lead ECGs to stratify for SCD risk compared to conventional ECG risk models. From the SUDS study in Portland Oregon, they identified 1827 out-of-hospital SCD ECGs from 1,796 cases. They validated the model in a separate cohort in Ventura County, CA, in 714 ECGs from 714 SCD cases. Two separate control group samples were obtained from 1,342 ECGs taken from 1,325 individuals of whom at least 50% had significant coronary

disease. The DL model was compared with a previously validated conventional 6-variable ECG risk model. The DL model achieved an AUROC of 0.889 (95% CI 0.861-0.917) for the detection of SCD cases vs. controls in the internal hold-out dataset, and was successfully validated in external SCD cases with an AUROC of 0.820 (0.794-0.847). The DL model performed significantly better than the conventional ECG model that achieved an AUROC of 0.712 (0.668- 38 0.756) in the internal and 0.743 (0.711-0.775) in the external cohort. They conclude that an ECG-based DL model distinguished SCD cases from controls with improved accuracy compared to a conventional ECG risk model.

This is a nice study.

- In the abstract and manuscript, it is incorrect to state “prospectively identified” since the analysis was run on already-collected data.

Reply: Thank you for your excellent and thoughtful comments. In both Oregon SUDS and Ventura PRESTO studies, all out-of-hospital SCD cases in the community are prospectively identified in collaboration with the region’s 2-tiered EMS system, regional hospital, and the state medical examiner’s office. We have described the methodology in more detail previously (PMID 15364331). However, pre-cardiac arrest ECGs were collected retrospectively after the SCD event. We have now clarified this in the Abstract (Page 2, line 11).

- Were all cases from SUDS and Ventura included – which if any were excluded?

Reply: Good question. As mentioned in the manuscript, all out-of-hospital SCD cases are prospectively identified in both Oregon SUDS and Ventura PRESTO studies. In this study, we excluded cases without prearrest ECG available for digitization, or if the prearrest ECG had paced rhythm, atrial fibrillation, or atrial flutter. We have now clarified the number of excluded cases in Figure 1.

- Were control cases propensity matched – on which variables?

Reply: Thank you for this comment. The controls were not propensity matched. We matched our controls based on the prevalence of coronary artery disease. The rationale for this is that the largest subgroup of SCD cases have significant coronary artery disease. Since our goal is to identify individuals at high risk of SCD (and not coronary disease) we matched the two groups on prevalence of coronary disease. We also performed a logistic regression analysis to evaluate the predictive power of the DL-ECG index beyond conventional clinical SCD risk factors, and findings indicate that the ECG-DL index is an independent predictor of SCD. Our rationale was that propensity matching is generally used to estimate the impact of an intervention in an observational setting. This study is observational, and not designed to investigate the impact of any intervention but to develop an ECG-DL model to identify individuals at high risk of SCD.

- **This is not clear – since training is usually completed prior to validation: “Based on the area under the curve of the 138 receiver operating characteristic (AUROC) curve in the validation dataset, we performed early 139 stopping for training”. Please clarify.**

Reply: Good question. The validation dataset in the internal cohort is used to evaluate the model during the training phase and prevent overfitting by performing early stopping of training. A separate testing dataset is used to evaluate the model performance, and all results in the internal cohort are derived from the testing dataset. We have illustrated this in Figure 1.

- **Please provide details of model dropout and architecture. “We limited the number of layers to less than 1/10th 142 the size of previously described architectures(20-22) to minimize 143 model complexity and optimize model runtime.” Please provide code as a supplement.**

Reply: Thank you for this comment. We have now provided a more detailed rationale for the model architecture in the Supplemental material. We have also provided the code as a supplement.

- **Please report predictive accuracy at different points in time.**

Reply: Thank you for this excellent comment. To clarify, our goal for this study was not to evaluate predictive accuracy at different points in time. This would be a particularly challenging prospect at the present since. To our knowledge sufficient numbers of ECGs are not available at different time points to conduct a feasible analysis, in any such study. In fact one of the unique aspects of our study is the overall availability of a feasible number of archived ECGs to conduct the present analysis. Accordingly, in the internal test dataset as well as the external cohort, only 1 prearrest ECG unrelated to the SCD event obtained at a single time point was utilized. The first step in our goal of long-term SCD risk stratification is to identify patients that are at an increased risk of SCD, and the ability to accurately predict when the event is going to occur is a future challenge. Identifying stable patients with a high risk of future SCD is the state-of-the-art practice in the prediction and prevention of SCD.

However, this is an excellent suggestion. In fact, we have a separate ongoing project where we are attempting to investigate the potential dynamic nature of SCD risk over time, but this does not involve deep learning analysis at the present time due to relatively small sample sizes.

- **Few ECG models are used to guide risk stratification beyond LVEF. It is not clear that a model that improves upon ECG index alone adds to models including LVEF. Is the ECG comparator useful?**

Reply: Thank you for this comment. We have performed logistic regression analyses to evaluate whether AI-ECG predicts SCD over and above potential confounding risk factors, including heart failure. Our analysis indicates that the AI-ECG index improved the discriminative value of SCD when added to a logistic regression model of clinical variables (from an AUC of 0.780 to 0.919 in the internal test dataset and from an AUC of 0.806 to 0.899 in the external validation dataset), demonstrating that the ECG-AI index has predictive power beyond conventional risk factors (Table 3,

Figure 4). We also recognize that these findings will need to be evaluated in prospective studies and clinical trials before implementation in the clinical setting.

- **Table 3. Please construct a model including SCD risk factors of LVEF and existing ECG risk factors e.g. QRS duration, history of AF – did DL add to predictions of such a model?**

Reply: Thank you for this comment. We have constructed SCD prediction models that include clinical SCD risk factors and conventional ECG abnormalities (Table 2, Table 3, Figure 3, Figure 4). The clinical SCD risk model included various clinical SCD risk factors (including heart failure, MI, etc), and the conventional ECG risk score is a cumulative sum of 6 ECG abnormalities: resting heart rate >75 bpm, LVH, delayed QRS transition zone ($\geq V5$), wide frontal QRS-T angle $>90^\circ$, prolonged Tpeak-to-Tend (>89 ms) interval, and prolonged QTc interval. We have compared the performance of our ECG-DL model to both the clinical SCD risk model and the conventional ECG-risk score and in both cases, the DL model achieved a higher accuracy in identifying SCD cases.

- **What is net reclassification rate using DL model compared to the clinically useful model above?**

Reply: Using a cutpoint of 0.70 for predicting case status, the net reclassification improvement using the DL model and clinical variables compared to the model with clinical variables in the internal cohort was 28.7% (95% CI 21.0% - 36.5%) and in the external cohort was 15.3% (95% CI 9.7% - 20.9%). We have now provided these findings in the Results (Page 10, Line 7).

- **Please report explainability of trained DL model – at least which regions of the ECG by Shapley or other metrics that most contributed to predictive value.**

Reply: Great suggestion. We have now used Local Interpretable Model-agnostic Explanations (LIME) to identify which ECG segments were used to identify SCD cases from controls. We have provided examples of LIME-highlighted ECG segments from two SCD cases and two control in the Supplement Figure. LIME highlighted a wide range of ECG features including PR interval, QRS complex, and ST- and T-wave changes. These findings are consistent with previous literature which has demonstrated that several ECG abnormalities are associated with an increased risk of SCD.

Reviewers' comments:

Reviewer #1 (Remarks to the Author):

The authors have done an excellent job addressing my concerns. I have no further comments.

Reviewer #2 (Remarks to the Author):

I have carefully reviewed the authors' responses to my comments on their manuscript regarding the deep learning model for predicting SCD risk from ECG data. I am satisfied with their clarifications and the adjustments made to the manuscript in line with my suggestions.

On Exclusion of Certain ECGs: The authors' justification for excluding ECGs with paced rhythm, atrial fibrillation, or atrial flutter is well-founded. Their clarification in the Methods section addresses the concern raised.

On Figure 1: I appreciate the authors' prompt incorporation of the number of excluded cases in Figure 1.

On Model Performance in Different Cohorts: The authors provided a comprehensive explanation for the performance variation between internal and external validation cohorts. Their discussion on the differences in demographics and clinical profiles between the cohorts and the implications of these differences on the model's generalizability is insightful.

On 2.5-second ECG strip: The authors' comment on the potential sufficiency of a 2.5-second ECG strip is valid given the results that they have demonstrated and their acknowledgment for further studies on this matter is appropriate.

On Patient Demographics: The authors' clarification that the external cohort improves generalizability is well taken.

The authors' explanation of how DL models can utilize the entire digital signal is concise and informative.

On Clinical Application of the Model: The authors' acknowledgment of the need for prospective studies and randomized trials is a prudent approach.

On Model Performance in Specific Categories: I appreciate the authors' thorough evaluation of the model's performance in different patient subgroups. The inclusion of these findings in the Results section is valuable.

On Biases and Limitations: The authors' suggestions for future studies to validate their model in prospective designs and diverse groups is a constructive approach to address potential biases.

Conclusion:

In light of the detailed responses provided by the authors and the modifications made to the

manuscript, I believe that the study has been strengthened. The authors have demonstrated a commendable effort in addressing the concerns raised during the review process.

Given the significance of the study's findings and its potential contributions to the cardiology field, I recommend the manuscript for publication in the journal.

Reviewer #3 (Remarks to the Author):

The authors have clarified most of the reviewer suggestions and substantially improved the manuscript. The work is done well, but a few design/conceptual limitations remain:

1. Control subjects were mismatched in ways that could inadvertently inflate the success of the model - they were healthier (younger, less diabetes, AF, CHF and COPD. As I mentioned before, propensity matching could have controlled for some of these variables.
2. Excluding ECGs with AF despite the fact that AF is a predictor of major adverse events and could plausibly precipitate ventricular arrhythmias via long-short intervals, in patients with pre-excitation (less likely) or via a pathophysiological disease association.
3. The same argument holds for pacing - although I do understand the rationale - which could plausibly be linked with SCD pathophysiology and/or trigger events by a mis-timed VP event.
4. The following statement is likely incorrect: "Including such ECGs may lead to a biased DL model that could learn to recognize paced rhythm, atrial fibrillation, or atrial flutter, instead of increased SCD risk." Not if the training label is SCD.
5. This statement is overly optimistic (Discussion): "... despite such differences, the overall performance remained good in the external cohort, which suggests good generalizability and mitigates the possibility of overfitting or systematic biases.." I may respectfully disagree given the design constraints above which make your population less representative than the general public.

Reviewers' comments:

Reviewer #1 (Remarks to the Author):

The authors have done an excellent job addressing my concerns. I have no further comments.

Response: Thank you!

Reviewer #2 (Remarks to the Author):

Conclusion: In light of the detailed responses provided by the authors and the modifications made to the manuscript, I believe that the study has been strengthened. The authors have demonstrated a commendable effort in addressing the concerns raised during the review process. Given the significance of the study's findings and its potential contributions to the cardiology field, I recommend the manuscript for publication in the journal.

Response: Thank you!

Reviewer #3 (Remarks to the Author):

The authors have clarified most of the reviewer suggestions and substantially improved the manuscript.

Response: Thank you!

The work is done well, but a few design/conceptual limitations remain:

1. Control subjects were mismatched in ways that could inadvertently inflate the success of the model - they were healthier (younger, less diabetes, AF, CHF and COPD. As I mentioned before, propensity matching could have controlled for some of these variables.

Response: We agree that propensity score matching is a good method to control for such variables. We opted to perform logistic regression model instead, so that we could control for these variables and also evaluate the predictive power of the DL-ECG index beyond conventional clinical SCD risk factors. However based on your comment we have included this aspect as a limitation as follows: "Although we aimed to match cases and controls based on the underlying CAD status, some differences in other SCD risk factors remained between SCD cases and controls, which may have affected the model performance". (Page 14 line 2)

2. Excluding ECGs with AF despite the fact that AF is a predictor of major adverse events and could plausibly precipitate ventricular arrhythmias via long-short intervals, in patients with pre-excitation (less likely) or via a pathophysiological disease association.

Response: We acknowledge that this is a limitation of our study imposed by the algorithm development process and have now included the following sentence on page 14, line 10: "Our model is only

applicable to sinus rhythm ECGs since atrial fibrillation/flutter and paced ECGs were excluded during algorithm development”.

3. The same argument holds for pacing - although I do understand the rationale - which could plausibly be linked with SCD pathophysiology and/or trigger events by a mis-timed VP event.

Response: We acknowledge that this is a limitation of our study imposed by the algorithm development process and have now included the following sentence on page 14, line 10: “Our model is only applicable to sinus rhythm ECGs since atrial fibrillation/flutter and paced ECGs were excluded during algorithm development”.

4. The following statement is likely incorrect: "Including such ECGs may lead to a biased DL model that could learn to recognize paced rhythm, atrial fibrillation, or atrial flutter, instead of increased SCD risk." Not if the training label is SCD.

Response: We have removed this statement from the manuscript.

5. This statement is overly optimistic (Discussion): "... despite such differences, the overall performance remained good in the external cohort, which suggests good generalizability and mitigates the possibility of overfitting or systematic biases.." I may respectfully disagree given the design constraints above which make your population less representative than the general public.

Response: We have removed the following words from this sentence: “...which suggests good generalizability and mitigates the possibility of overfitting or systematic biases” (Pg 10 line 18).

REVIEWERS' COMMENTS:

Reviewer #3 (Remarks to the Author):

The authors have addressed my comments.